# Effect of Ultra-High Pressure Homogenisation (UHPH) on the Co-Inoculation of *Lachancea thermotolerans* and *Metschnikowia pulcherrima* in Tempranillo Must

**DOI:** 10.3390/biom14121498

**Published:** 2024-11-24

**Authors:** Cristian Vaquero, Carlos Escott, Iris Loira, Carmen López, Carmen González, Juan Manuel Del Fresno, Buenaventura Guamis, Antonio Morata

**Affiliations:** 1Madrid Culinary Campus (MACC), Universidad Pontificia de Comillas, Calle Alberto Aguilera 23, 28015 Madrid, Spain; 2Departamento de Farmacia Galénica y Tecnología Alimentaria, Sección Departamental de Farmacia Galénica y Tecnología Alimentaria, Facultad de Veterinaria, Universidad Complutense de Madrid, Avenida Puerta de Hierro s/n, 28040 Madrid, Spain; carlos.escott@ucm.es; 3enotecUPM, Escuela Técnica Superior de Ingería Agronómica, Alimentaria y de Biosistemas (ETSIAAB), Universidad Politécnica de Madrid, Avenida Puerta de Hierro 2, 28040 Madrid, Spain; iris.loira@upm.es (I.L.); carmen.lopez@upm.es (C.L.); carmen.gchamorro@upm.es (C.G.);juanmanuel.delfresno@upm.es (J.M.D.F.); antonio.morata@upm.es (A.M.); 4Centre d’Innovació, Recerca i Transferència en Tecnologia dels Aliments (CIRTTA), TECNIO, XaRTA, Departament de Ciència Animal i dels Aliments, Facultat de Veterinària, Universitat Autònoma de Barcelona, 08193 Bellaterra, Spain; buenaventura.guamis@uab.cat

**Keywords:** wine, sequential, non-*Saccharomyces*, biocompatibility, yeast nutrients, non-thermal treatment, ternary fermentations

## Abstract

The utilisation of non-*Saccharomyces* yeasts in co-inoculation and non-thermal technologies for must sterilisation is becoming increasingly prevalent due to their notable utility and potential. This new approach optimises the fermentation process and contributes to facilitating the production of wines with distinctive characteristics, improving their stability, and without organoleptic repercussions. Two trials were conducted concurrently, designated as A and B, using the same Tempranillo red must. In each trial, UHPH-treated and untreated must (serving as the control) were compared. The non-*Saccharomyces* yeasts (*Lachancea thermotolerans* and *Metschnikowia pulcherrima*) were identical in both trials, and fermentations were terminated by a *Saccharomyces cerevisiae* inoculated after 7 days (ternary fermentation). In Trial A, different percentages of the initial inoculum were employed with respect to the total volume that must be fermented, with the objective of evaluating the influence and competitiveness between yeasts. Trial B was designed to investigate the impact of two nutrients that provide vitamins, energy, and protection from oxidative stress on the development of these yeasts and their metabolic expression. Microbiological analysis and measurements of oenological parameters were carried out, acidification was assessed, volatile compounds were analysed, and the colour spectrum was measured by spectrophotometry. It was observed in both trials that the prevalence of *Lachancea thermotolerans* (Lt) was longer than that of *Metschnikowia pulcherrima* (Mp) and that the use of quercetin + thiamine had a positive effect on yeast growth. Furthermore, the combination of Lt and Mp yeasts demonstrated remarkable synergy, resulting in the production of a substantial quantity of lactic acid (>5 g/L). With regard to aroma compounds, the UHPH must have exhibited a nearly twofold increase in ethyl lactate. Additionally, the total polyphenol index (TPI) was observed to be 8–10% higher in wines derived from UHPH musts, indicating that this technology may potentially safeguard against oxidation.

## 1. Introduction

Today, the use of non-*Saccharomyces* yeasts in winemaking is increasingly widespread due to their high diversity and, therefore, their novel metabolic profiles that differentiate wines [1,2]. These distinctive attributes can facilitate the enhancement of relatively flat grape profiles, thereby yielding wines that are both expressive and well-received by consumers [3]. Furthermore, the potential for co-inoculation with non-*Saccharomyces* yeasts introduces the possibility of generating interesting synergies at the oenological level [4,5]. Currently, the most studied mixed and sequential inoculations are those of the genera *Lachancea*, *Metschnikowia*, *Hanseniaspora,* and *Torulaspora* [6,7]. The yeast species *Lachancea thermotolerans* can produce up to 10–11% alcohol and generate up to 12 g/L of lactic acid through the metabolization of sugars. Additionally, it increases ethyl lactate and lowers pH, rendering it an appealing yeast for use in quality wine production where wines tend to remain at high pH [8,9]. The yeast species *Metschnikowia pulcherrima* can produce a high volatile acidity (>0.7 g/L) [10]. Furthermore, yeast produces pulcherriminic acid, which, depending on the amount synthesised, can have an antimicrobial effect [11]. However, it is also a yeast that is used as a starter due to its ability to generate highly distinctive esters, thiols, and terpenes, including pear-scented ethyl octanoate and other aromas strongly associated with fermentation, such as 2,6-dimethoxyphenol, which evokes smoked foods [12,13].

It has been studied that yeast fermentation kinetics are directly related to the type and amount of nutrients in the medium and that yeast growth may be compromised if any nutrient is lacking. Quercetin is a very abundant flavonoid that has been shown to protect yeasts from oxidative stress by activating certain proteins through its pro-oxidant properties [14,15]. Thiamine, or vitamin B1, has an essential cofactor role for several enzymes involved in various metabolic pathways, including those leading to the production of wine-relevant aroma compounds, and it also supports yeast survival through stress protection functions [16,17].

The use of ternary fermentations with three or more yeasts is becoming increasingly prevalent to fully harness the expressive potential of wines [5,18,19]. The aforementioned yeasts serve as illustrative examples of starter yeasts that can enhance the organoleptic characteristics of wines [20]. Non-*Saccharomyces* yeasts are not particularly competitive and demonstrate limited resistance to SO_2_; therefore, the utilisation of non-thermal technologies, such as high hydrostatic pressure (HHP), pulsed electric fields (PEF), pulsed light (PL), and ultra-high-pressure homogenisation (UHPH), is a promising approach to reduce or eliminate the initial microbial load in the must, thereby facilitating the successful implantation of the desired yeasts [21,22,23,24]. In addition to the elimination of microorganisms, UHPH technology inactivates oxidative enzymes (polyphenol oxidase (PPOs)) by generating nanofragmentation due to the continuous pressurisation of the fluid to be used at a pressure exceeding 200 MPa; this process occurs without appreciable organoleptic changes [21,25]. On the other hand, it has been studied that, like HHP technology, UHPH produces a slight impact on both the colour and terpenes of fruit juices [26,27]. It should be added that 5-methyl furfural and 5-hydroxymethylfurfural have not been detected in the musts processed with UHPH; that is, this process does not thermally degrade pentoses and hexoses [21,28]. The aim of this work was to analyse the biocompatibility, evolution, and metabolite production of the following yeasts in ternary fermentations (co-inoculations), in all cases finishing the fermentation sequentially with *Saccharomyces cerevisiae* for the production of high-quality wines in warm areas: *L. thermotolerans* and *M. pulcherrima*, with a *S. cerevisiae* strain as a control and with two different conditions, Trial A, with different volumes of initial inoculum and Trial B, with the use of two nutrients, all with must untreated by UHPH and with must treated by UHPH.

## 2. Materials and Methods

### 2.1. Ultra-High Pressure Homogenisation of Musts

A pneumatic press was used to crush the *Vitis vinifera* L. Tempranillo grapes at a temperature of 15 °C. The resulting must was then separated into two parts; one part was treated by UHPH, and the other part was used as a control. The Tempranillo skinless must was processed using a continuous UHPH system (60 L/h), at an inlet temperature of 23–25 °C and outlet temperature of 13–15 °C, valve temperature of 78–85 °C for only 0.02 s, and flow rate of 60 L/h at 300 ± 3 MPa. The total volume processed by the UHPH was 60 L [26], patented by the Universitat Autònoma de Barcelona and manufactured by Ypsicon Advanced Technologies in Barcelona, Spain (EP2409583B1).

### 2.2. Yeast Strains and Growing Media

The yeast strains used in this experimental set-up were all isolated at the Food Technology Laboratory of the Escuela Técnica Superior de Ingeniería Agronómica, Alimentaria y de Biosistemas (Universidad Politécnica de Madrid). The starter species used were two non-*Saccharomyces* yeasts in co-inoculation (*Lachancea thermotolerans* (Lt) strain L3.1 [9,29] and *Metschnikowia pulcherrima* (Mp) strain M29 [30]). All vinifications were completed with sequential fermentation by inoculating the *Saccharomyces cerevisiae* (Sc) yeast strain 7VA on the seventh day after the start of fermentation to completely exhaust the sugars present in the must.

To achieve a high and similar population for the fermentation trials, all strains were grown for 24 h in a liquid YEPD medium at a constant temperature of 24 °C. The liquid medium for yeast growth was prepared by combining 1% yeast extract, 2% bacteriological peptone, and 2% anhydrous D(+)-glucose from Panreac Química of Barcelona, Spain (Laboratorios Conda, Madrid, Spain). The culture medium was autoclaved at 120 °C for 15 min.

### 2.3. Micro-Fermentation Trials

As shown in Figure 1, the microfermentation trials were performed in ISO flasks of 100 mL capacity with 80 mL of must *Vitis vinifera* L. cv. Tempranillo with ~245 g/L of sugars and a pH of ~3.7. In total, two different musts were used, one of which was Tempranillo without any treatment and another must that was treated by UHPH according to the instructions in Section 2.1. Two trials were carried out at the same time with these yeasts and the same must but with different conditions. In Trial A, both non-*Saccharomyces* strains were inoculated in equal parts (50:50), but the volume of the initial inoculum was 2% (Ca and Ua) and 0.2% (Cb and Ub), to study the influence of UHPH on the implantation and biocompatibility of different inoculum volumes. In Trial B, both non-*Saccharomyces* strains were inoculated in equal parts (50:50) and at 2% inoculum volume, but quercetin (1 g/L, “Cq” and “Uq”) was added to one batch and quercetin + thiamine (1 g/L + 0.6 mg/L, respectively, “Cqt” and “Uqt”) was added to the other batch to evaluate the use of these fermentation activators on the metabolic performance of the co-inoculated non-*Saccharomyces* species. All fermentations were carried out in triplicate and continued for 25 days until sugar exhaustion (<2 g/L). In all the vinifications, sequential fermentation with Sc took place on day 7 by adding 2% of the precultured strain 7 VA. After completion of fermentations, five millilitres of each wine triplicate were filtered through 0.45 ηm MCE syringe membranes (Branchia, Dismadel, Madrid, Spain) for analytical characterisation.

### 2.4. Yeast Population Counts

Inoculated yeast populations were measured by plating serial dilutions on CHROMagar™ Candida (Conda, Barcelona, Spain). Samples were taken from the fermentations on days 0, 3, and 6, and *Saccharomyces* was inoculated on day 7. With this selective medium, a rigorous count was achieved in the co-inoculations between *L. thermotolerans* and *M. pulcherrima*, as their colour and shape had already been identified in previous investigations [5,31].

### 2.5. Oenological Parameters

General oenological parameters were determined in the grape juices and finished wines with the use of FTIR spectrometry (FOSS, Barcelona, Spain) and with the use of a Y25 enzymatic analyser (Byosystems, Barcelona, Spain). The analysis comprised the determination of amino nitrogen, ammonia, total sugars, organic acids (tartaric acid and malic acid) in grape juices, while organic acids (lactic acid (by Y25), acetic acid), residual sugars (glucose and fructose), ethanol, and pH were measured in finished wines. Additionally, pH values were measured with a Crison Instruments GLP 21 (Hach Lange Spain, S.L.U., Madrid, Spain). Five mL samples were filtered with a 0.45 ηm membrane, and the trapped CO_2_ was removed with agitation.

### 2.6. Aroma Volatile Compounds

Volatile compounds were analysed in accordance with a previously described method [26,32]. The identification of volatile compounds was carried out with an Agilent Technologies™ 6850 chromatograph (Palo Alto, CA, USA) with a column DB-624 (60 m × 250 μm × 1.4 μm). The GC-FID parameters were as follows: injector’s temperature was 250 °C; detector’s temperature was 300 °C; and hydrogen was the carrying gas with a flow of 2.2 L/min and split ratio of 1:10. Finally, the temperature increased from 40 °C for 5 min to 250 °C with a gradient of 10 °C/min and was maintained for 5 min. The identification and the quantification of volatile organic compounds were performed on 1 mL of previously filtered samples and the use of individual calibration curves for the following: 2-phenylethyl acetate, 2-phenylethanol, ethyl acetate, isobutyl acetate, ethyl butyrate, isoamyl acetate, acetaldehyde, methanol, 1-propanol, diacetyl, 1-butanol, 2-butanol, isobutanol, acetoin, 2-methyl-1-butanol, 3-methyl-1-butanol, ethyl lactate, 2,3-butanediol, and 1-hexanol. Lastly, in accordance with the procedure previously described, 100 μL of 4-methyl-2-pentanol (500 mg/L; Fluka Chemie GmbH, Buchs, Switzerland) was used as the internal standard. The detection limit was 0.1 mg/L. The volatile compounds analysed with this technique were precalibrated with five-point calibration curves (r^2^), and all compounds had an r^2^ > 0.999, except 2,3-butanediol (0.991) and phenylethyl alcohol (0.994).

### 2.7. Colour Parameters

The absorbance at 280, 420, 520, and 620 nm was determined at the end of fermentation using an Agilent 8453 spectrophotometer (Agilent Technologies S.L., Madrid, Spain) and a 1 mm pitch quartz cuvette. To comply with the standard method, the result was multiplied by 10 to emulate the optical passage through a 1 cm cuvette, and parameters such as TPI, colour intensity, and tonality were determined.

### 2.8. Statistics

ANOVA and the least significant difference (LSD) test were used to analyse the differences and determine the means and standard deviations. PCA and all calculations were performed with the program PC Statgraphics v.5 (Graphics Software Systems, Rockville, MD, USA). The significance was set to *p* < 0.05.

## 3. Results and Discussion

Two studies were carried out to look at the mixed implantation of non-*Saccharomyces* yeasts, on the one hand, with the use of different volumes of starter inoculum, and on the other hand, with the use of different nutrients. All these used the same must Tempranillo, but one part of the must had been treated with UHPH and the other not, in order to verify the effect of this sanitisation technology on the metabolic development of the yeasts used as starters. Non-thermal technologies such as the one used in these trials, in addition to pulsed light (PL) and pulsed electric fields (PEFs), among others, are increasingly used in the field of oenology to partially or totally eliminate wild microflora from musts without causing organoleptic changes and thus be able to inoculate the desired yeasts with a higher success rate [22,23,33].

### 3.1. Evolution of the Inoculated Population

The yeast population and its evolution during fermentation (Figure 2) were monitored until day 6 through the CHROMagar™ Candida culture medium, just before the inoculation of *S. cerevisiae* on day 7. In Trial A (Figure 2A), the initial inocula exhibited a 10-fold difference, with one (6-log CFU/mL, 2% initial inoculum) being higher than the other (5-log CFU/mL, 0.2% initial inoculum), which was as anticipated. On the third day, both yeasts demonstrated reduced growth in the UHPH-treated musts. In both cases, the UHPH-treated musts exhibited no indigenous yeasts at the outset or throughout fermentation because the colours, textures, and shapes of the colonies were easily recognised with the selective medium used. By the sixth day, the yeast populations had decreased overall by 1-log CFU/mL. The prevalence of *L. thermotolerans* was observed to persist for a longer period than that of *M. pulcherrima*. From day 3 onwards, the viability of the latter declined at a more rapid rate, which is likely attributable to its poor ethanol tolerance [6]. In general, the exponential growth of *M. pulcherrima* was observed to be lower, which is likely attributed to its relatively low competitiveness [34]. Furthermore, the use of UHPH-treated musts resulted in a reduction in yeast population growth, which is likely attributable to a decline in nutrient bioavailability and subsequent utilisation [30].

On the other hand, in Trial B (Figure 2B), an initial inoculation at 2% inoculum was performed, as the initial population was sufficient to ensure rapid and consistent growth of the inoculated yeast, ensuring that exponential growth phases were reached in a reasonable time [21,35], and it was observed that yeast growth was similar, reaching more than 7-log in *L. thermotolerans* on the third day and almost 7-log on average in *M. pulcherrima*, results expected due to the synergy seen between different non-saccharomyces in previous work, which anticipated that fermentations would be exponential to 7-log until day 3–4 and then gradually declining in population [5,36]. Furthermore, it was observed that the combination of quercetin and thiamine resulted in an average increase of 0.5 log in population size compared to that observed in the presence of quercetin alone. For both yeast strains, a decline in population was observed from the sixth day of fermentation, similar to that observed in Trial A; only indigenous yeasts were present in the non-UHPH-treated controls. The combination of quercetin + thiamine appears to exert a beneficial effect on yeast growth in both control and UHPH must, potentially due to its capacity to safeguard yeast cells from oxidative stress [15,16]. The antimicrobial efficacy of UHPH was unmistakably demonstrated by the absence of indigenous yeast growth [21,26]. One compound to consider is pulcherriminic acid, which can be generated by *M. pulcherrima* and pose a risk in ternary fermentations such as these. The antifungal effect of this acid can worsen the implantation and growth of L3.1 as it depletes iron (III) in the form of a ferric complex known to inhibit the proliferation of other species [37,38].

### 3.2. Oenological Parameters

The initial must have a sugar concentration of approximately 245 g/L. In Trial A, it was observed that the fermentation process was slower in the musts with UHPH, resulting in the accumulation of residual sugars at the end of fermentation (Table 1). This phenomenon can be attributed to the reduced availability of certain nitrogenous compounds that may be nano-encapsulated and less accessible [26]. Nevertheless, the greatest quantity of alcohol was achieved with the UHPH musts. With regard to total acidity, the highest levels were observed in the fermentation with control musts (without UHPH) and at high inoculum doses, with a value of 8.27 ± 0.26 g/L in tartaric acid equivalents. Volatile acidity was found to be below 0.5 g/L in all cases.

In contrast, the fermentation process in Trial B (Table 2) was more homogeneous, although residual sugars below 0.5 g/L were present in the fermentations with UHPH must. In all cases, the volatile acidity was below 0.5 g/L.

It was observed that fermentations with higher total acidity, both in Trial A and especially in Trial B (there being a difference in total acidity of more than 4 g/L from initial to final [29]), exhibited a reduction in ethanol production at the end of fermentation, reaching a level of approximately 1% *v/v* less than expected (~14.5% *v*/*v*). This phenomenon can be attributed to the high lactic acid production of *L. thermotolerans*, which utilises a portion of the sugars present in the must to generate this elevated acidity [39]. Furthermore, although not the subject of investigation in these trials, numerous non-*Saccharomyces* yeasts have the capacity to produce pyruvic acid, glycerol, and other compounds from sugars through secondary metabolic processes [40]. It should be noted that certain Mp strains have the capacity to reduce the final ethanol content by up to 1.6% *v/v* when co-inoculated with a Sc [41]. In accordance with the characteristics of mixed fermentation, the volatile acidity was found to be relatively low, with a concentration below 0.5 g/L in both trials, a finding that aligns with the observations of other researchers in this field [30,42].

### 3.3. Lactic Acidity and pH

It has been demonstrated that Lt produces the highest amount of lactic acid between days 3 and 6 of fermentation [9]. However, this process is contingent upon a number of variables, including temperature, nutrients, and the competitive dynamics between yeast strains. Consequently, these factors must be considered in any analysis [5,43]. To check the effect of bioacidification with lactic acid on pH, we compared these results in Trial A (Figure 3A) and saw a high correlation (>−0.9), where the control sample with high initial inoculum (2%) produced a higher amount of lactic acid (4.30 ± 0.09 g/L) and, therefore, a lower pH, 3.20 ± 0.01. On the other hand, in Trial B (Figure 3B), the production of lactic acid was more pronounced, especially in the must sample with UHPH and the nutrient quercetin, producing 6.18 ± 0.34 g/L lactic acid and a pH of 3.10 ± 0.03.

With these results, it was observed that nutrients can generate a higher production of lactic acid compared to the average of Ca and Ua samples, producing 1.5 g/L more lactic acid in Trial B. Additionally, the observed synergy between Lt and Mp yeasts was found to be highly effective without the occurrence of incompatibilities, as previously documented in other studies [5,21,38].

It should be added that there is a close relationship between yeast count and lactic acid production. Previous work has shown that a population above or around 7-log CFU/mL is needed to initiate significant lactic acid formation and that this amount of yeast is sufficient to contribute to the sensory profile of wines, especially in terms of freshness, through lactic acid production [9,44]. In addition, this increase in acidity also provides advantages by the consequent increase in molecular SO_2_ obtained at lower pH, thus allowing the reduction of total SO_2_ with the same stability, having a bioprotective effect, and influencing the production of off-flavours [45,46].

### 3.4. Colour Assessment

The absorbances of both trials were evaluated spectrophotometrically at different wavelengths to determine, after fermentation, their colour intensity, hue, and total polyphenol index (TPI). In Trial A (Table 3), a considerably higher colour intensity than the rest was observed in the control sample with the high initial inoculum (Ca), probably because its pH is lower and the anthocyanins are predominantly found in their protonated form and their colour is more reddish [47]. It is also noteworthy that both the hue and the IPT of the control samples were significantly lower than in the wines obtained from the UHPH-treated musts.

The same evaluation was conducted with Trial B, and it was observed that the controls with both quercetin and quercetin + thiamine exhibited higher colour intensity at the conclusion of fermentation than the UHPH. Furthermore, as observed in Trial A, the hue and IPT were significantly lower in the control wines than in the UHPH.

Higher IPT in wines from UHPH musts means a higher concentration of these compounds, and this technology protects from oxidation [21]. On the other hand, quercetin can undergo hydrolysis of the glycosyl bond during fermentation, releasing the quercetin aglycone, which is much less soluble in aqueous solution and leads to precipitation and, as seen in our assay, a 5% reduction of total polyphenols compared to Trial A [48]. It was also calculated that the samples that had produced more lactic acid in both Trials A and B (Cq, Uq, Cqt, Uqt, Uqt, and Ca) generated about 9% less yellow hues (420 nm) [30,38].

### 3.5. Aromatic Profile

The volatile compounds that are produced during the fermentation process of wine play an important role in determining the wine’s sensory profile; the inoculated yeasts are a significant factor in this process, influencing the types and quantities of aromas that are synthesised [49,50]. The volatile compounds of fermentation origin were subjected to analysis in both trials. In Trial A (Figure 4A), a high initial inoculum (Ca and Ua) produced lower amounts of higher alcohols and total esters. Furthermore, carbonyl compounds were generally low. Notably, for esters, excluding ethyl acetate, it was observed that UHPH wines contained almost twice as much ethyl lactate. Conversely, Trial B revealed that the simultaneous addition of quercetin and thiamine resulted in the production of elevated levels of higher alcohols in the UHPH wines, while the control wine exhibited enhanced formation of total esters (not including ethyl acetate) and carbonyl compounds.

In all cases, the higher alcohols were present in concentrations above 300 mg/L, which could impart subtle spicy and fusel oil aromas [51]. It should be noted that the alcohol that exerted the most pronounced effect was 3-methyl-1-butanol (isoamyl alcohol), which produces complex, smoky, and roasted aromas when its concentration exceeds 30 mg/L [52,53]. The second most abundant alcohol was 1-propanol, with a mean of 67 mg/L. However, this was considerably below the perceived threshold of 830 mg/L [54]. This discrepancy can be attributed to the limited scale of the trial, as at industrial levels, this alcohol does not typically emerge as a prominent factor [9,42]. The concentration of carbonyl compounds, which impart notes of candy and butter, was below the detection threshold for diacetyl (4–12 mg/L) and acetoin (150 mg/L) in both trials [55,56]. It can be stated that esters are aromas related to the amount of nitrogen (NFA) that the yeast is able to assimilate [54]. Ethyl acetate is a compound that, at concentrations below 60 mg/L, exhibits pleasant fruity aromas. In all fermentations, its concentration remained below this threshold, with an average of ~48 mg/L. It was observed that in one of the fermentations (Cb), the production of ethyl acetate was considerably higher (~85 mg/L); this has been shown to be due to the fact that the same fermentation had a higher population of *M. pulcherrima* producing this volatile and could have produced even more if it had not been co-inoculated [57]. Ethyl lactate, whose production is a direct consequence of lactic acid production by Lt [29], also remained below the perception threshold of 150 mg/L, with an average concentration of ~43 mg/L [29,58]. With an average yield of less than 1.5 mg/L in all ethyl butyrate fermentations, it is a volatile that produces fruity aromas, especially apples, with a threshold of perception at 0.02 mg/L [59,60]. Both isoamyl acetate with sweet banana and fruit aroma with a hint of ripe essence and isobutyl acetate with tropical fruit aroma, especially banana, were detected in several samples, but especially in the control sample with quercetin + thiamine with ~4 mg/L for both volatiles, which could be detectable as their perception threshold is below 2 mg/L for both volatiles [54,56]. 2-phenylethyl acetate is a volatile compound characteristic of *Hanseniaspora* spp. with floral (rose) odours, although the amounts of this compound were low, with an average of 6 mg/L but always above the perception threshold of 0.25 mg/L, which together with the other volatile compounds produces a high complexity of aromas [53,61]. It should be noted that, although not depicted in the figures, acetaldehyde was also analysed as a volatile compound, yielding values approximately 20 times higher in the control wines (approximately 300 mg/L). These elevated values may be attributed to a number of factors, including the composition of the medium, the nature of the insoluble materials employed to clarify the musts, the ageing/breeding procedure, the SO₂ content, the aeration, and most notably, the indigenous yeast population, which differs significantly between the control must and the UHPH must. It is highly probable that a population of *S. cerevisiae beticus* was present [62]. Additionally, the presence of non-*Saccharomyces* species, including *C. stellata*, *Z. bailii*, and *S. pombe* [63], as well as bacteria such as *Lactobacillus plantarum*, may have contributed to the observed fermentation profile [64].

## 4. Conclusions

The use of UHPH eliminates the microflora naturally present in the grape must, facilitating the implantation of yeast ferments and preventing spontaneous fermentations from taking place. Moreover, the use of non-*Saccharomyces* yeasts can be strongly favoured in both mono- and co-inoculation. It should also be noted that both the nutrients (quercetin + thiamine) and the non-*Saccharomyces* yeasts studied produced some very interesting synergies that open up a very powerful field of study to be explored further.

## Figures and Tables

**Figure 1 biomolecules-14-01498-f001:**
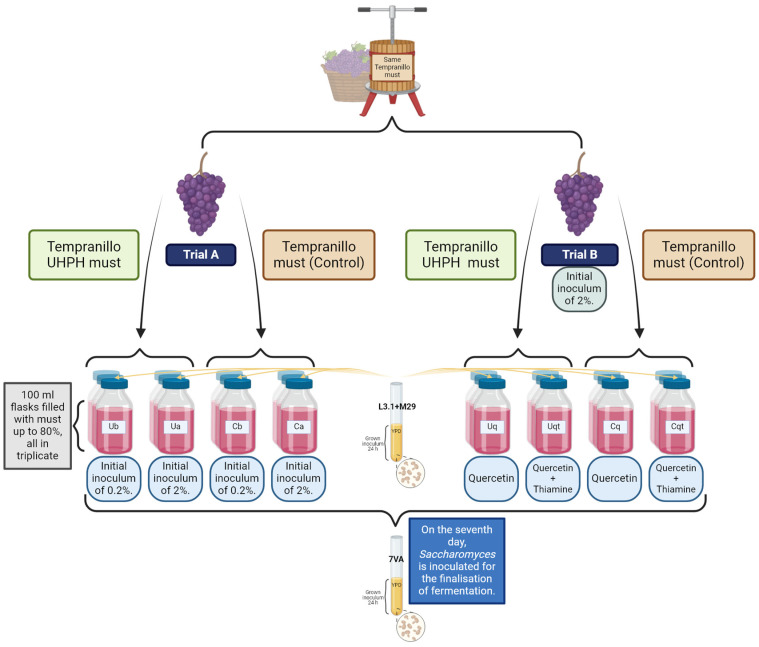
Schematic representation of the experimental process for the evaluation of the co-inoculation of non-*Saccharomyces* under different inoculation volumes and nutrient addition, and the use of UHPH-treated or untreated must. “U” = with UHPH treatment, “C” = control without UHPH treatment, “B” = 0.2% of initial inoculum, “A” = 2% of initial inoculum, “Q” = quercetin, “QT” = quercetin + thiamine.

**Figure 2 biomolecules-14-01498-f002:**
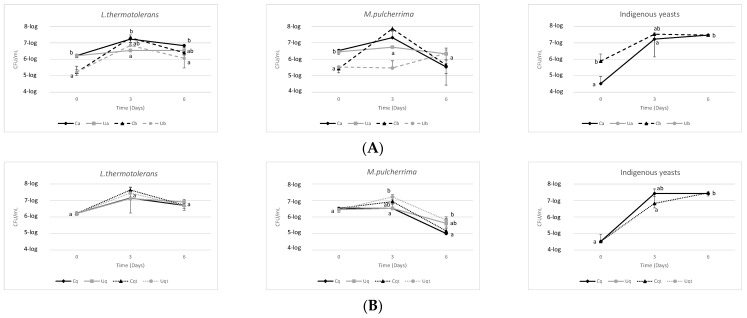
(**A**) Evolution of yeast populations in Trial A with two different inoculum volumes (a = 2% inoculation volume, b = 0.2% inoculation volume, C = Control must, and U = UHPH treated must). (**B**) Evolution of the yeast populations in Trial B with the addition of quercetin and quercetin + thiamine. Values are means ± SD (n = 3). A different letter for the same day means significant differences (*p* < 0.05).

**Figure 3 biomolecules-14-01498-f003:**
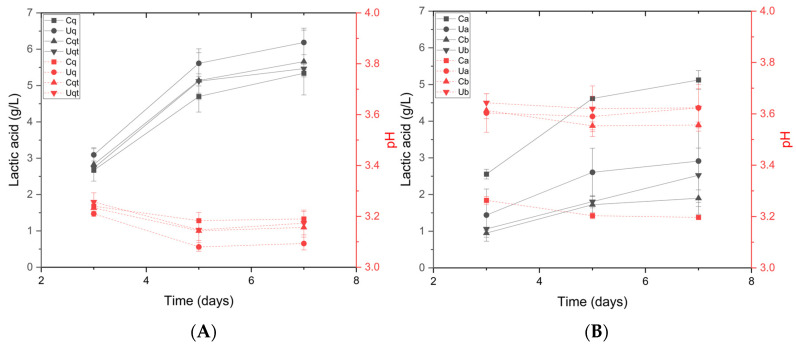
(**A**) Lactic acid production (solid line) versus pH decrease (dashed line) in Trial A. (**B**) Lactic acid production (solid line) versus pH decrease (dashed line) in Trial B. a = 2% inoculation volume, b = 0.2% inoculation volume, C = control must, and U = UHPH treated must.

**Figure 4 biomolecules-14-01498-f004:**
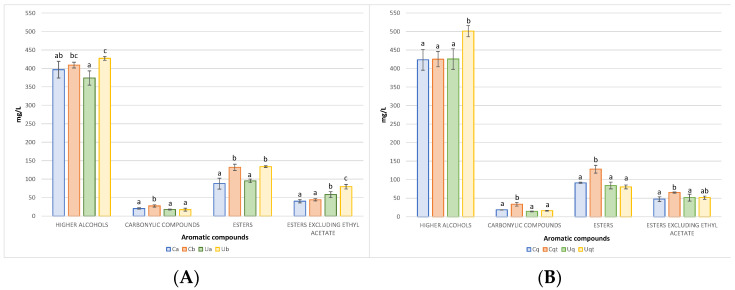
(**A**) Aromatic compounds from Trial A. (**B**) Aromatic compounds from Trial B. Values are means ± SD (n = 3). A different letter for the same compound group means significant differences (*p* < 0.05).

**Table 1 biomolecules-14-01498-t001:** Oenological parameters of the first trial with different initial inoculum volumes. Different letters in each column indicate statistical differences (*p* < 0.05) between treatments in each fermentation scenario.

	Residual Sugars (g/L)		Ethanol (% *v*/*v*)		Total Acidity ^1^ (g/L)		Volatile Acidity ^2^ (g/L)	
Trial/Days	3	25	3	25	3	25	3	25
Ca	154.40 ± 2.23 a	0.00 ± 0.00 a	4.07 ± 0.15 c	12.25 ± 0.32 a	7.65 ± 0.03 c	8.27 ± 0.26 c	0.35 ± 0.01 c	0.44 ± 0.05 c
Ua	208.77 ± 6.43 c	2.07 ± 0.90 b	2.03 ± 0.15 a	14.09 ± 0.15 c	5.15 ± 0.46 b	6.41 ± 0.49 b	0.30 ± 0.02 b	0.29 ± 0.02 b
Cb	170.77 ± 3.10 b	0.00 ± 0.00 a	3.70 ± 0.26 b	13.52 ± 0.34 b	5.21 ± 0.09 b	5.30 ± 0.11 a	0.28 ± 0.02 ab	0.20 ± 0.02 a
Ub	216.30 ± 2.08 d	1.87 ± 0.49 b	1.93 ± 0.06 a	14.54 ± 0.15 c	4.30 ± 0.24 a	6.10 ± 0.52 b	0.26 ± 0.02 a	0.25 ± 0.06 ab

^1^ Expressed as tartaric acid. ^2^ Expressed as acetic acid.

**Table 2 biomolecules-14-01498-t002:** Oenological parameters of the second trial with quercetin and quercetin + thiamine nutrients. Different letters in each column indicate statistical differences (*p* < 0.05) between treatments in each fermentation scenario.

	Residual Sugar (g/L)		Ethanol (% *v*/*v*)		Total Acidity ^1^ (g/L)		Volatile Acidity ^2^ (g/L)	
Trial/Days	3	25	3	25	3	25	3	25
Cq	154.43 ± 4.31 a	0.00 ± 0.00 a	4.17 ± 0.23 a	12.89 ± 0.17 a	7.71 ± 0.13 c	8.36 ± 0.24 a	0.35 ± 0.01 a	0.39 ± 0.03 bc
Uq	172.13 ± 4.88 a	0.30 ± 0.17 b	3.60 ± 0.26 a	13.56 ± 0.85 a	7.32 ± 0.13 b	9.20 ± 0.18 b	0.35 ± 0.02 a	0.30 ± 0.02 a
Cqt	163.93 ± 18.89 a	0.00 ± 0.00 a	3.67 ± 0.93 a	13.04 ± 0.26 a	7.71 ± 0.13 c	8.52 ± 0.31 a	0.34 ± 0.02 a	0.46 ± 0.07 c
Uqt	168.37 ± 2.47 a	0.47 ± 0.25 b	3.90 ± 0.10 a	13.56 ± 0.06 a	7.07 ± 0.07 a	8.74 ± 0.13 a	0.33 ± 0.02 a	0.32 ± 0.03 ab

^1^ Expressed as tartaric acid. ^2^ Expressed as acetic acid.

**Table 3 biomolecules-14-01498-t003:** Trial A and Trial B parameters of colour intensity, tonality, and total polyphenol index (TPI) were measured on the last day of fermentation (day 25). Values are means ± sd (n = 3). A different letter for the same parameter and assay means significant differences (*p* < 0.05).

		Colour Intensity ^1^ (Absorbance Units)	Tonality ^2^ (Adimensional)	Total Polyphenol Index (Absorbance Units)
Trial A	Ca	2.13 ± 0.01 b	6.46 ± 0.01 a	15.23 ± 0.25 a
	Ua	1.78 ± 0.09 a	10.26 ± 0.10 c	17.19 ± 0.72 b
	Cb	1.64 ± 0.01 a	7.63 ± 0.01 b	15.76 ± 0.15 a
	Ub	1.74 ± 0.01 a	10.39 ± 0.01 c	17.06 ± 0.51 b
Trial B	Cq	2.00 ± 0.05 b	6.53 ± 0.05 a	14.95 ± 0.15 a
	Uq	1.48 ± 0.05 a	10.22 ± 0.06 b	16.03 ± 0.64 ab
	Cqt	2.10 ± 0.03 b	6.52 ± 0.04 a	15.10 ± 0.15 a
	Uqt	1.56 ± 0.10 a	10.79 ± 0.13 b	16.40 ± 1.00 b

^1^ Σ of absorption at λ_max_420 nm, λ_max_520 nm and λ_max_620 nm. ^2^ Ratio λ_max_420 nm/λ_max_520 nm.

## Data Availability

A detailed explanation of all procedures is given in the materials and methods, and should any researcher require further details, please write to the principal author.

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
