# Peer review of "Effect of Ultra-High Pressure Homogenisation (UHPH) on the Co-Inoculation of Lachancea thermotolerans and Metschnikowia pulcherrima in Tempranillo Must"

_biomolecules, 2024, doi:10.3390/biom14121498_

Round 1
Reviewer 1 Report
Comments and Suggestions for Authors
Review report
This article proposes a study of the use of non-Saccharomyces yeasts in co-inoculation, combined with non-thermal technologies for must sterilization.
The article summarizes the results obtained. However, certain details should be included to deepen the manuscript and facilitate understanding.
1.Introduction
The authors could provide a clearer explanation of how UHPH works and highlight its relevance in oenology for microbiological stabilization. Additionally, they should elaborate on the reasoning behind exploring the addition of nutrients to the medium and why these two specific nutrients were chosen.
2. M&M
The authors could provide more details on the basic analyses of the initial must.
Clarify whether the UHPH treatment was applied to wine with or without grape skins.
In Figure 1, it would be helpful to label the legends consistently with the other figures, using uniform formatting.
Additionally, specify the range of 4-methylpentanol.
3. Results
The authors never mention filtration. Why? Filtration is a conventional, non-thermal microbiological stabilization strategy and is likely less energy-intensive than UHPH.
Figure 2: Standardize the axes, scales, and colors.
The authors should discuss the results in more depth. Additional discussion on the different parameters studied could be provided.
The authors do not consider the possibility of modulating color with UHPH. It is also used as an extraction technology.
Figure 4: Format with the same scale and color code.
Table 3: Clarify the legend within the table.

Author Response
Review report
-This article proposes a study of the use of non-Saccharomyces yeasts in co-inoculation, combined with non-thermal technologies for must sterilization. The article summarizes the results obtained. However, certain details should be included to deepen the manuscript and facilitate understanding.
-Thank you very much for appreciating our work. We have improved the manuscript according to your comments.
1.Introduction
-The authors could provide a clearer explanation of how UHPH works and highlight its relevance in oenology for microbiological stabilization. Additionally, they should elaborate on the reasoning behind exploring the addition of nutrients to the medium and why these two specific nutrients were chosen.
-We have added an additional paragraph in the introduction to go into more depth on the topic of nutrients and we have also added more information on the great utility of UHPH technology, all supported by references.
- M&M
-The authors could provide more details on the basic analyses of the initial must.
-Throughout the materials and methods we have added more relevant information, including an additional section (2.4) to complete the information.
-Clarify whether the UHPH treatment was applied to wine with or without grape skins.
-We have made a clarification in section 2.1 of M&M.
-In Figure 1, it would be helpful to label the legends consistently with the other figures, using uniform formatting.
-We have increased the information on the image of figure 1 and added more information to the front of the figure for better compression.
-Additionally, specify the range of 4-methylpentanol.
-We have added more information on the volatiles analysis methodology in M&M part 2.6.
- Results
-The authors never mention filtration. Why? Filtration is a conventional, non-thermal microbiological stabilization strategy and is likely less energy-intensive than UHPH.
-Thank you very much for your question. In the paper we do not mention filtration because it is the most used technique in the winery and we decided to study and use other technologies that are more advanced than filtration, because for example the UHPH technology in addition to sterilizing the must, inactivates the polyphenol oxidase enzymes and we do not have appreciable organoleptic losses. Moreover, it is a continuous and fast process without the need to change any filter or filter plate. In any case, the initial must be filtered (not amicrobically) so that no particles larger than 0.5 mm remain. (https://doi.org/10.1007/s11947-022-02766-8)
-Figure 2: Standardize the axes, scales, and colors.
-Figure 4: Format with the same scale and color code.
-Table 3: Clarify the legend within the table.
-We have unified colours, scales and sizes in the different figures to avoid problems in their visualization and improved the table you mention.
-The authors should discuss the results in more depth. Additional discussion on the different parameters studied could be provided.
- We have added several comments to better and more deeply discuss the results obtained.
-The authors do not consider the possibility of modulating color with UHPH. It is also used as an extraction technology.
- Totally agree with your comment, in our previous studies we observed a slightly higher concentration of anthocyanins in the UHPH musts, especially due to the selective protection of the UHPH in the acylated derivatives. In this essay we decided not to go that deep, probably in future trials we will take it into account again and analyze it in more depth. Thank you very much for your comment.
-We sent the article to a specialist website where it was edited to ensure that it followed British English conventions with appropriate structure and vocabulary.
Reviewer 2 Report
Comments and Suggestions for Authors
In the manuscript “Effect of ultra high pressure homogenisation (UHPH) on the co- inoculation of Lachancea thermotolerans and Metschnikowia pulcherrima in Tempranillo must with the use of different nutrients and inoculum concentrations” authors present a study regarding the effect of UHPH in the Tempranillo wine made with a pre-inoculation of non-sacharomyces yeasts prior to the S.cerevisiae inoculation. The article is interesting and fits within the scope of the journal. This reviewer considers that the following items should be improved:
Major comments:
Abstract: Many factors are included as object of investigation: UHPH, coinoculation with two yeast strains at different inoculum concentration and different nutrients. Maybe it should be better if authors focalize in fewer items (at least in the title) and introduce lates (in the abstract) that they, besides, evaluated different inoculum concentrations, i.e.
Moreover, the two trials are not comparable. And maybe they should not be put as in parallel. Or at least, in the Figure 1, the inoculum of yeasts in trial B should be stated. Trial 2 was made with an unique concentration of inoculum (2%). In Figure 1 caption, the meaning of UB, UA, CB, CA, UQ, UQT….etc should be stated in order to facilitate the readers comprehension.
Figure 2: how did authors quantified the “indigenous yeasts”? Moreover: how they counted all the yeasts? Author mention in the result section that they used the CHROMagar™ Candida culture medium, but no mention of this counting was made in the M&M section. Please, add this information in M&M section.
The preference of using 2% of initial inoculum of non-saccharomyces yeasts instead of 0,2% in Trial B should be mentioned somewhere in the discussion section.
Minor comments
Line 23: “its notable utility and potential”…for what?
Line 24-25: “This approach optimises (….) facilitates (….)”?
or “This approach investigates (or aim) the optimization (….) and contribute (or aim) the facilitation of…”.
Line 51: please put the name of the species in italics.
Figure 4B: there is some kind of “shadow” in the basis on the bars.
Reference 14: the name of the authors is in capital letters. Please put them in the same format than the others
Comments on the Quality of English LanguageMinor details were found.
Author Response
-In the manuscript “Effect of ultra high pressure homogenisation (UHPH) on the co- inoculation of Lachancea thermotolerans and Metschnikowia pulcherrima in Tempranillo must with the use of different nutrients and inoculum concentrations” authors present a study regarding the effect of UHPH in the Tempranillo wine made with a pre-inoculation of non-sacharomyces yeasts prior to the S.cerevisiae inoculation. The article is interesting and fits within the scope of the journal. This reviewer considers that the following items should be improved:
-Thank you very much for appreciating our work. We have improved the manuscript according to your comments.
Major comments:
-Abstract: Many factors are included as object of investigation: UHPH, coinoculation with two yeast strains at different inoculum concentration and different nutrients. Maybe it should be better if authors focalize in fewer items (at least in the title) and introduce lates (in the abstract) that they, besides, evaluated different inoculum concentrations, i.e.
-We have shortened the title and then expanded and deepened the information in the abstract.
-Moreover, the two trials are not comparable. And maybe they should not be put as in parallel. Or at least, in the Figure 1, the inoculum of yeasts in trial B should be stated. Trial 2 was made with an unique concentration of inoculum (2%). In Figure 1 caption, the meaning of UB, UA, CB, CA, UQ, UQT….etc should be stated in order to facilitate the readers comprehension.
- We have improved Figure 1 by adding a caption with more details and within the figure we have added the initial percentage of inoculum taking into account your comments.
-Figure 2: how did authors quantified the “indigenous yeasts”? Moreover: how they counted all the yeasts? Author mention in the result section that they used the CHROMagar™ Candida culture medium, but no mention of this counting was made in the M&M section. Please, add this information in M&M section.
-We have added an additional section (2.4) in the Materials and Methods to clarify the use of the CHROMagar™ Candida selective medium and how this selective medium can tell us what type of yeast is growing, and we have also included several references to support our experience in using this selective medium.
-The preference of using 2% of initial inoculum of non-saccharomyces yeasts instead of 0,2% in Trial B should be mentioned somewhere in the discussion section.
-We have added a clarification of the use of the initial inoculum percentage in the second paragraph of section 3.1, following your comment.
Minor comments
-Line 23: “its notable utility and potential”…for what? Line 24-25: “This approach optimises (….) facilitates (….)”? or “This approach investigates (or aim) the optimization (….) and contribute (or aim) the facilitation of…”.
-We have made several changes and clarifications in the abstract for better understanding.
-Line 51: please put the name of the species in italics.
-Done, we have written it in italics.
-Figure 4B: there is some kind of “shadow” in the basis on the bars.
- We have unified colors and sizes in the different figures to avoid problems in their visualization.
-Reference 14: the name of the authors is in capital letters. Please put them in the same format than the others
- Done, we have rewritten it correctly.
-We sent the article to a specialist website where it was edited to ensure that it followed British English conventions with appropriate structure and vocabulary.
Reviewer 3 Report
Comments and Suggestions for Authors
The authors report an interesting work on the potential of high pressures to promote non-Saccharomyces dominance in scalar ternary inoculum designs. Please highlight that the title is intended to promote dominance.
In the abstract, please ensure that the use of S. cerevisiae is also clear. Improve the number of keywords.
I suggest to improve the introduction in order to highlight the different aspects and elements of novelty associated with the study (e.g. the strains have been inoculated in these combinations before, evidence from previous studies with other strains, use of other pressures to improve the dominance of other strains, the study on nutrition/nutrients, the use of this specific must/grape variety). Please improve the introduction and, consequently, the rest of the manuscript. Please ensure that appropriate literature is cited (e.g. line 59 The use of ternary fermentations with three or more [...] (10.3390/fermentation6020055), 10.3389/fmicb.2021.656262, 10.3390/fermentation7030171)). Please ensure that species names are always in italics (e.g. line 51 Lachancea thermotolerans).
Concerning materials and methods, please, i) provide references if the strains used have been previously characterized and used for oenology, ii) ensure that adequate literature is cited for each methodological aspect.
Please increase the size of the figures. When revising the introduction, please improve the detail of the results trying to highlight all the novel aspects and to adequately discuss all these aspects.
Comments on the Quality of English LanguageYou used both North American and British words in your manuscript. Please unify the use of British words as much as possible.
Please revise the text to improve clarity (e.g. lines 32-33 "Microbiological analysis and oenological parameter measurements, acidification assessments, volatile compound analyses, and spectrophotometric evaluations were conducted")
Author Response
-The authors report an interesting work on the potential of high pressures to promote non-Saccharomyces dominance in scalar ternary inoculum designs. Please highlight that the title is intended to promote dominance.
-Thank you for your comment. We have shortened the title to better highlight the key aspects of the paper.
-In the abstract, please ensure that the use of S. cerevisiae is also clear. Improve the number of keywords.
-We have included a clarification in the abstract regarding the use of Saccharomyces yeast, and we have improved and expanded the keywords according to your comments.
I suggest to improve the introduction in order to highlight the different aspects and elements of novelty associated with the study (e.g. the strains have been inoculated in these combinations before, evidence from previous studies with other strains, use of other pressures to improve the dominance of other strains, the study on nutrition/nutrients, the use of this specific must/grape variety). Please improve the introduction and, consequently, the rest of the manuscript. Please ensure that appropriate literature is cited (e.g. line 59 The use of ternary fermentations with three or more [...] (10.3390/fermentation6020055), 10.3389/fmicb.2021.656262, 10.3390/fermentation7030171)). Please ensure that species names are always in italics (e.g. line 51 Lachancea thermotolerans).
-We have added new information to expand on the issues discussed in the paper, added the papers you mentioned, and corrected the lack of italics.
-Concerning materials and methods, please, i) provide references if the strains used have been previously characterized and used for oenology, ii) ensure that adequate literature is cited for each methodological aspect.
-Thank you very much for the comment. We have added references for the strains used, as well as additional citations throughout the materials and methods section to ensure clarity.
-Please increase the size of the figures. When revising the introduction, please improve the detail of the results trying to highlight all the novel aspects and to adequately discuss all these aspects.
-We have enlarged the images to the maximum possible extent reaching the margins. On the other hand, we have improved the introduction by delving into different aspects.
-Comments on the Quality of English Language. You used both North American and British words in your manuscript. Please unify the use of British words as much as possible.
-We sent the article to a specialist website where it was edited to ensure that it followed British English conventions with appropriate structure and vocabulary.
-Please revise the text to improve clarity (e.g. lines 32-33 "Microbiological analysis and oenological parameter measurements, acidification assessments, volatile compound analyses, and spectrophotometric evaluations were conducted")
-Thank you for your comment. We have expanded the abstract by providing clearer and more in-depth information.
Round 2
Reviewer 2 Report
Comments and Suggestions for Authors
Authors addressed this reviewer comments. Article is suitable for publication.
Author Response
Thank you very much for your careful review of the manuscript.
Reviewer 3 Report
Comments and Suggestions for Authors
The authors addressed some of the criticisms.
- Lines 26-28: please consider modifying this part to improve clarity. E.g. "Two trials were conducted concurrently, designated as A and B, using the same Tempranillo red must. In each trial, UHPH treated and untreated must (serving as the control) were compared. The non-Saccharomyces yeasts (Lachancea thermotolerans and Metschnikowia pulcherrima) were identical in both trials, and fermentations were terminated by a Saccharomyces cerevisiae inoculated after 7 days (ternary fermentation)."
- In the abstract is missing a part on the "Effect of UHPH on the co-inoculation of Lachancea thermotolerans and Metschnikowia pulcherrima in Tempranillo must"
- I suggest to standardize all the codes by comparing figure 1 and the other graphs/tables. Eg. in Fig. 1 you used UA, UB, CA, CB, in the other graphical elements you used Ua, Ub, Ca, Cb
- In figure 2, on the y-axis, the numbers and the logarithmic representation are superimposed. Please check and improve the figure.
- Please ameliorate the discussion in all the sections, trying to improve the impact of your findings considering all the points (e.g. what's new about UHPH Tempranillo must?, what's new about UHPH and non-saccha dominance?, what's new about co-inoculation of two non-sacchas in Tempranillo?, what's new about ternary fermentations with Saccha incoculation at 7 days?, how to discuss lactic data in light of microbial count data, ...?). If needed, please improve the abstract/conclusions accordingly.
- In figures 3 and 4 the letters A and B partially cover the figures.
Comments on the Quality of English LanguageThe quality of English is good in my opinion. After the scientific reviews, I can suggest small changes.
Author Response
The authors addressed some of the criticisms.
- Lines 26-28: please consider modifying this part to improve clarity. E.g. "Two trials were conducted concurrently, designated as A and B, using the same Tempranillo red must. In each trial, UHPH treated and untreated must (serving as the control) were compared. The non-Saccharomyces yeasts (Lachancea thermotolerans and Metschnikowia pulcherrima) were identical in both trials, and fermentations were terminated by a Saccharomyces cerevisiae inoculated after 7 days (ternary fermentation)."
- Thank you very much for your input, the change has been made as suggested.
- In the abstract is missing a part on the "Effect of UHPH on the co-inoculation of Lachancea thermotolerans and Metschnikowia pulcherrima in Tempranillo must"
-We consider that with the new approach you suggest in the first point, plus all the information in the abstract, it is possible to conclude that it clearly alludes to the title, and furthermore, putting more information in the abstract would go far beyond the word limit that the journal requires of us.
- I suggest to standardize all the codes by comparing figure 1 and the other graphs/tables. Eg. in Fig. 1 you used UA, UB, CA, CB, in the other graphical elements you used Ua, Ub, Ca, Cb
- Figure 1 has been changed, standardising all codes.
- In figure 2, on the y-axis, the numbers and the logarithmic representation are superimposed. Please check and improve the figure.
-It has been checked and the logarithms are in line with the CFU/ml of each yeast on the different days tested.
- Please ameliorate the discussion in all the sections, trying to improve the impact of your findings considering all the points (e.g. what's new about UHPH Tempranillo must?, what's new about UHPH and non-saccha dominance?, what's new about co-inoculation of two non-sacchas in Tempranillo?, what's new about ternary fermentations with Saccha incoculation at 7 days?, how to discuss lactic data in light of microbial count data, ...?). If needed, please improve the abstract/conclusions accordingly.
-Several sections of the discussion have been expanded and deepened on different points.
- In figures 3 and 4 the letters A and B partially cover the figures.
- In figures 3 and 4 the letters have been moved so that they do not cover any relevant information.
Thank you very much for your careful review of the paper.
Round 3
Reviewer 3 Report
Comments and Suggestions for Authors
The authors addressed the main criticisms
Comments on the Quality of English LanguagePlease make a final revision of the text